# EPR and Related Magnetic Resonance Imaging Techniques in Cancer Research

**DOI:** 10.3390/metabo13010069

**Published:** 2023-01-01

**Authors:** Yoichi Takakusagi, Ryoma Kobayashi, Keita Saito, Shun Kishimoto, Murali C. Krishna, Ramachandran Murugesan, Ken-ichiro Matsumoto

**Affiliations:** 1Quantum Hyperpolarized MRI Research Team, Institute for Quantum Life Science, Quantum Life and Medical Science Directorate, National Institutes for Quantum Science and Technology, 4-9-1 Anagawa, Inage-Ku, Chiba 263-8555, Japan; 2Department of Quantum Life Science, Graduate School of Science, Chiba University, Chiba 265-8522, Japan; 3Radiation Biology Branch, Center for Cancer Research, National Cancer Institute, National Institutes of Health, Bethesda, MD 20892-1002, USA; 4Karpaga Vinayaga Institute of Medical Sciences and Research Center, Palayanoor (PO), Chengalpattu 603308, India; 5Quantitative RedOx Sensing Group, Department of Radiation Regulatory Science Research, National Institute of Radiological Sciences, Quantum Life and Medical Science Directorate, National Institutes for Quantum Science and Technology, 4-9-1 Anagawa, Inage-Ku, Chiba 263-8555, Japan

**Keywords:** electron paramagnetic resonance imaging, electron spin resonance imaging, redox imaging, spectral–spatial imaging, oxygen mapping, Overhauser-enhanced MRI, dynamic nuclear polarization, hyperpolarization, metabolic imaging, ^13^C-labeled pyruvate

## Abstract

Imaging tumor microenvironments such as hypoxia, oxygenation, redox status, and/or glycolytic metabolism in tissues/cells is useful for diagnostic and prognostic purposes. New imaging modalities are under development for imaging various aspects of tumor microenvironments. Electron Paramagnetic Resonance Imaging (EPRI) though similar to NMR/MRI is unique in its ability to provide quantitative images of pO_2_ in vivo. The short electron spin relaxation times have been posing formidable challenge to the technology development for clinical application. With the availability of the narrow line width trityl compounds, pulsed EPR imaging techniques were developed for pO_2_ imaging. EPRI visualizes the exogenously administered spin probes/contrast agents and hence lacks the complementary morphological information. Dynamic nuclear polarization (DNP), a phenomenon that transfers the high electron spin polarization to the surrounding nuclear spins (^1^H and ^13^C) opened new capabilities in molecular imaging. DNP of ^13^C nuclei is utilized in metabolic imaging of 13C-labeled compounds by imaging specific enzyme kinetics. In this article, imaging strategies mapping physiologic and metabolic aspects in vivo are reviewed within the framework of their application in cancer research, highlighting the potential and challenges of each of them.

## 1. Introduction

Altered cellular metabolism is well recognized as one of the stellar hallmarks of cancer. Recent technological advances have facilitated deeper understanding of the complex changes in carbohydrate, amino acid, and lipid metabolism and thereby provide opportunities for targeted molecular imaging and therapeutic interventions. Several functional imaging modalities to detect pathophysiological information of tumor are recently developed to aid cancer research [1,2,3,4,5,6,7]. An appropriate therapeutic treatment can be planned based on the prior information derived from functional imaging. In addition, the efficacy of the treatment could be verified based on the information observed in the follow-up functional imaging. Such interrelationship between diagnosis and treatment, mediated by the functional imaging techniques have evolved into the theranostics applications in cancer research.

Functional imaging techniques, which are essential for achieving theranostics, can visualize cellular function in the target tissues/organs. Visualization of pharmacodynamics of ^18^F-labeled deoxy glucose (FDG) using positron emission tomography (FDG-PET) is a representative example of functional imaging that is widely utilized in the clinic [8,9,10]. The tissue with high glucose metabolism, especially cancer tissues, can be detected by the FDG-PET. A major example of functional imaging in magnetic resonance modalities is the blood-oxygen level dependent (BOLD) MRI, which detects differences of paramagnetic properties in the tissue caused by ratio-shift of paramagnetic deoxy-hemoglobin to diamagnetic oxy-hemoglobin during a sequential T_2_-weighted image acquisition [11,12]. The BOLD MRI technique can evaluate responses of brain neuronal activation to specific tasks or in the resting state by estimating change in blood oxygenation.

Electron paramagnetic resonance (EPR) has many technical aspects in common with nuclear magnetic resonance (NMR). NMR detects absorption of radio frequency by nuclear spins in an applied magnetic field, while EPR detects conventionally the absorption of microwave frequency by unpaired electron spins in a magnetic field. For better tissue penetration, low frequency EPR techniques have been developed. Proton MRI can provide clear anatomical images of internal organs/tissues and is widely used in clinic for diagnostic purpose. In contrast, EPR imaging maps the distribution of stable free radicals used as the contrast agent and provides physicochemical information surrounding the free radical. In this respect, EPRI is similar to PET imaging which requires positron-emission isotope-labeled agent. Thus, EPR imaging is advantageous in providing functional information of cells/tissues/organs rather than morphological/anatomical information. EPR is currently preclinically used for investigating tumor models in rodents [13,14]. Specifically, tissue redox status [15,16,17] and tissue oxygen level are the major targets of CW and pulsed EPR imaging modalities [18,19,20,21].

Dynamic nuclear polarization (DNP) was widely used to enhance the intrinsically low-sensitive NMR signals. A paramagnetic contrast agent with narrow EPR line width is used for polarization transfer from the electron spin to the surrounding nuclear spin with the consequent NMR signal enhancement. The principle of DNP was discovered by Overhauser in 1953 [22]. Hence, this double resonance technique is called Overhauser-enhanced MRI (OMRI). This technique is known by other names such as Proton Electron Double Resonance Imaging (PEDRI) and Electron Spin Resonance-Enhanced MRI (ESREMRI). OMRI/PEDRI provided MRI-based highly resolved anatomical information, in addition to the EPR spectroscopic functional information [23,24]. Application of OMRI to cancer research was illustrated by acquiring functional (pO_2_) images co-registered with tumor morphology [25,26]. Other applications of OMRI/PEDRI include separation of multiple free radical species in a sample [27] and tissue redox status estimation [28,29,30].

Next stage of DNP application in cancer research was set to extend this double resonance technique for metabolic imaging, based on a hyperpolarized ^13^C-labeled compound [31]. The DNP of ^13^C at liquid helium temperature can enhance ^13^C signal to noise ratio (SNR) by several orders of magnitude. The hyperpolarized ^13^C-labeled compound is immediately injected to the subject after dissolution. The altered metabolism can be evaluated by analyzing ^13^C NMR signals at different chemical shifts, corresponding to the different metabolites in spectral imaging. Energy production in cancerous tissue shifts to glycolysis rather than TCA cycle. Such metabolic shift can be translated into upregulated pyruvate to lactate flux in the hyperpolarized ^13^C pyruvate MRI.

Hypoxia is a most popularly investigated feature in cancer [32,33,34]. Hypoxia alters cancer cell metabolism and plays a pivotal role in cancer progression and dissemination. It induces cell quiescence and results in therapy resistance. The tumor tissue redox status can also be modified by intracellular conditions, such as hypoxia, pH, GSH level, ascorbate level, mitochondrial ROS generations, and/or mitochondrial metabolic activities. Tumor tissue shows its redox change from relatively early stage and it looks synchronized with tumor volume and oxygen level [35]. Therefore, sensing the tissue oxygen levels and tissue redox status would be helpful to finding the tumor/cancer at its earlier stage.

In addition, tissue hypoxia and the attending redox status is important factors for planning and evaluating the efficacy of radiation therapy. Hypoxic cancer tissues are radio-resistant. Compared to anoxic conditions, well-oxygenated environment shows greater efficacy to therapy which is known as the oxygen effect of radiation. Oxygen-induced enhancement of radio-biological effects are significantly higher for X-ray or γ-ray as compared to high linear energy transfer (LET) particle beam such as α-ray, proton beam, carbon beam, or other heavy-ion beams. Much higher LET particle beam, such as iron-ion beam, even showed low level of oxygen induced enhancement of radio-biological effects [36]. Tissue oxygen level, which is generally quite low even in a normal tissue, could modify radiation chemistry in the tissue. Generally, 80% of the effect of ionizing radiation originates as a result of indirect actions, in which water molecules are ionized and several reactive species are generated. Such reactive species can interact with molecular oxygen to produce superoxide, hydrogen peroxide, and other reactive oxygen species, which eventually form hydroxyl radical. Redox processes in the tissues could be enzymatically and chemically modified by endogenous antioxidative enzymes and antioxidants. Oxygen concentration, redox status, and other metabolic activities in the target tissue are therefore important parameters for planning a radiation therapy [37,38,39].

Intratumor oxygen concentration measurement, prior to the treatment, is of significant importance in planning suitable therapy and in predicting the outcome of the therapy [40]. PET can sense hypoxic regions in tissues with the use of positron-emitting radioisotope-labeled hypoxic markers, such as ^18^F-FMISO, ^64^Cu-ATSM, etc. BOLD and tissue oxygen level dependent (TOLD) MRI techniques can also provide tissue oxygenation level. However, such imaging techniques are susceptible to local blood flow and are not capable of quantitatively evaluating oxygen concentration in the tissue. On the other hand, EPR-based oximetry is a quantitative technique for oxygen concentration mapping, independent of perfusion. Redox imaging, which visualizes the reduction in exogenously administered nitroxyl radical can estimate such conditions of the target tissue. EPR imaging [41], T_1_-weighted MRI [42] and OMRI/PEDRI [43] are applied to visualize tissue redox status, by using stable nitroxide radicals as redox probes.

For diagnostic as well as therapeutic application, the information of pO_2_, redox and metabolic status in both cancer and normal tissue would be useful for not only finding and/or diagnosing the focus but also supporting radiation therapy of cancer [5,6]. In this review, developmental background and current status of EPR spectroscopy and imaging techniques as well as preclinical utility of EPR-related imaging modalities such OMRI and hyperpolarized MRI for cancer research are described.

## 2. Development of Functional EPR Imaging Techniques

Earlier EPR imaging experiments used CW method and filtered back-projection reconstruction algorithms were used to obtain the images from the spatially encoded EPR spectra, collected in the presence of static magnetic field gradients. The earliest EPR imaging experiments were reported by two groups almost simultaneously in 1979 [44,45], which was 6 years after the development of MRI by Lauterbur [46]. Earlier applications of EPR imaging were in the fields of material science [47,48,49]. Meanwhile, pulsed MRI technique was developed in the 1975 [50], following which the first commercial clinical MRI scanner was made available in 1980 [51]. With the big success of MRI as a medical imaging device, development of in vivo EPR imaging became an active area of research.

### 2.1. In Vivo EPR Imaging

Initially gradient coil modules were incorporated in the commercially available X-band EPR spectrometer for imaging. X-band EPR spectrometer operates at a frequency around 9.4 GHz. Aqueous samples suffer large dielectric loss due to absorption of radiation by water molecules at this frequency, inducing heat in irradiated objects [52]. This dielectric loss limits the available volume of imaging in X-band EPR measurement to be less than 100 μL. To avoid the heating problem, low frequency EPR such as L-band (500–1500 MHz) or radio frequency (<300 MHz) was employed for preclinical in vivo EPR imaging [53,54]. Several types of resonators such as surface coil [55], parallel coil [56], loop-gap [57], re-entrant resonators [58], etc., were introduced for low frequency in vivo EPR measurements. Resonator design which can minimize dielectric loss and give high magnetic flux in the active volume continues to be of interest in EPR imaging.

The first L-band EPR imaging experiment was reported by Berliner and Fujii [59], and the spatially encoded EPR spectra representing 1-dimensional distribution of a stable nitroxyl radical, TEMPOL, absorbed into a celery stem were obtained using a magnetic field gradient of 24.3 Gauss/cm. The first in vivo 2D EPR imaging in an animal was also reported by Berliner et al. [60]. Distribution of TEMPOL in a melanoma, grown in a live mouse tail, was observed. Following the success of these initial studies, several groups have developed in vivo EPR imaging instrumentation for visualizing redox probes such as nitroxyl radical, in experimental animals [61,62,63].

Schematic presentation of encoding spatial distributions of a stable free radical in the CW EPR spectrum using a static magnetic field gradient is given in (Figure 1). When a free radical probe gives a single narrow line EPR spectrum, the spectrum obtained under a field gradient can be directly used as projection data of 1D spatial distribution profile for reconstructing the EPR image. Rotation of the field gradient can give a projection data set to obtain a 2D or 3D EPR image (Figure 2). However, EPR spectral information, such as linewidth and hyperfine splitting can result in artifact of pseudo spatial distribution. To minimize this artifact, the EPR spectral information must be eliminated from the EPR spectrum by using deconvolution process. Ideally, use of single narrow line spin probe as the imaging agent can minimize these problems and enhance the image resolution.

Distribution of a paramagnetic contrast agent provides limited information for a biomedical application. Application of EPR imaging to cancer research can be achieved by attaching an additional dimension to the distribution map of the paramagnetic contrast agent. The temporal axis can be easily attached by repeating measurement of images as a function of time [13,64]. Consequently, an EPR signal decay rate mapping, which is utilized in redox mapping, can be obtained from a set of temporal EPR images. EPR spectroscopic characteristics, however, can give additional biologically important information, such as pO_2_, pH, and micro-viscosity, based on variation of EPR spectral parameters. Analysis of overloading such functional information in EPR image data has been is reported at the very early stage of EPR imaging technique development [65,66]. In addition, power saturation behavior, which is related with relaxation time of electron spin, also utilized for EPR oxygen mapping [67].

### 2.2. Pulsed EPR Imaging

Electron spin-echo-detected spectral–spatial EPR imaging technique was reported following the initial CW EPR imaging development. Two different radical species located at different positions were imaged using spin echoes collected after a 180°-τ-90° pulse sequence [68,69]. It was reported almost simultaneously with CW EPR spectral–spatial imaging at X-band [70,71]. In principle, the pulse techniques already developed in MRI could be extended to EPR imaging. However, the extremely short relaxation time of electron spins (~ nano’s time scale) poses great challenge in developing instrumentation for using pulsed magnetic field gradient for slice selection as in MRI. Hence, for spectral–spatial 2D EPR imaging, filtered back-projection image reconstruction using a set of projections, obtained by Fourier transformation of the FID or the echo signals observed under static field gradient was employed [72,73]. Almost 10 years after the first in vivo CW EPR imaging, the first in vivo application of the pulsed 300 MHz EPRI, based on FID-detection and projection reconstruction was reported in 1997 [74]. Following the success, low frequency (radio frequency) in vivo pulsed EPR imaging experiments based on back-projection reconstruction were attempted [75,76,77,78,79]. Single-point imaging (SPI) modality was first introduced in pulsed EPR imaging using static gradients. In this modality, the FIDs are encoded in k-space for Fourier image reconstruction [80] (Details are provided in the next section). EPR spectral–spatial imaging technique, using free induction decays (FIDs) and SPI image reconstruction algorithm was developed as an effective method for electron spin T_2_^*^ based oximetric imaging [81]. On the other hand, Halpern et al. have reported an electron T_1_ based oximetric imaging by electron spin echo detection modality [82,83].

### 2.3. Spectral–Spatial EPR Imaging (Oxygen Mapping)

In 2D EPR imaging, one of the spatial dimensions can be replaced by spectral dimension, resulting in the so called the spectral–spatial EPR imaging. The spectral dimension can be used for oxygen mapping, based on the pO_2_ dependent linewidth changes. In 1986, Ewert and Herrling reported 2D EPR spectral–spatial imaging [84]. The projection reconstruction modality of the spectral–spatial imaging method is basically similar to the initial NMR spectral–spatial imaging [85,86,87]. Theory of EPR spectral–spatial image reconstruction algorithms is well summarized by Maltempo [71] who developed the basic principles of EPR spectral–spatial imaging technique [70,71]. A 3D EPR spectral–spatial imaging system at 1.3 GHz was developed for oximetry of an isolated rat heart and rabbit aorta using TEMPO as a contrast agent [88]. Four-dimensional EPR spectral–spatial images of phantoms and an isolated rat heart were obtained using ^15^N-d_16_-TEMPONE as contrast agent [89]. The main purpose of developing in vivo EPR spectral–spatial imaging technique was to acquire oximetric imaging. Another application of EPR spectral–spatial imaging technique was for the separation of multiple free radical species in a single experimental system [90,91].

The ground state of molecular oxygen, which has two unpaired electrons, is a triplet sate with spin angular momentum of 1, 0, and −1. However, direct detection EPR resonance of the paramagnetic oxygen by conventional X-band EPR at room temperature is extremely difficult because the relaxation time of molecular oxygen is too short and the EPR spectrum of molecular oxygen is very broad (spans from 200 to nearly 1300 mT). Nevertheless, oxygen concentration in the sample can be derived by observing EPR line broadening of stable paramagnetic (free radical) probe. As a result of spin-spin interaction with molecular oxygen the relaxation time of the free radical probe would be shortened. Using a stable free radical compound with narrow EPR linewidth, such as the triarylmethyl radical, LiPc, etc., as oximetry probe, both CW and pulsed EPR spectral–spatial imaging can provide in vivo and in vitro oxygen mapping. The single-point imaging (SPI) modality introduced in FID-based pulse EPR imaging [80] has a few advantages over the projection-based reconstruction. The SPI modality practically eliminates the linewidth effects and resonator dead-time effects, which can result distortion in spatial and spectral distribution when FID based projection reconstruction technique is applied. The SPI modality, also referred to as constant-time imaging (CTI), was first introduced in NMR imaging of solids, [92,93]. The SPI modality for EPR image reconstruction from the FIDs is summarized in Figure 3 and Figure 4.

A set of FIDs observed with a series of incremented field gradient gives a sequential SPI data at each time point after the pulse (Figure 3) which can be used for Fourier transformed constant-time spectral–spatial imaging (FT-CTSSI) [94]. For oxygen mapping, T_2_* which is the decay rate of FID (the slope of exponential decay of FID) can be estimated from the reconstructed time axis of CTSSI data. From 2D, 3D, or 4D CTSSI data, 1D, 2D, or 3D spatial distribution of T_2_* of the stable free radical (oxygen probe) can be computed (Figure 3 and Figure 4). The T_2_* or linewidth values would be translated to the oxygen concentration using a standard calibration curve acquired from data of the spin probe solutions equilibrated at various oxygen concentrations.

The spatial resolution of SSI is varied depending on the time after the RF pulse, τ. Short τ gives large FOV and low spatial resolution. Long τ gives small FOV and high spatial resolution. When the T_2_* is estimated using multiple time points on the reconstructed FID, lower spatial resolution on shorter τ gives pseudo value on the edge of the subject. As a result, the large edge artifacts in T_2_^*^/linewidth mapping can be observed [94]. To overcome this problem, several different maximum field gradient settings were used to acquire multiple data set of CTSSI to set the FOV of all SPI images of different τ almost identical. Such SPI set with a constant spatial resolution was re-assembled (Figure 4, lower panel) and 3D oxygen mapping in tumor inoculated in mouse hindleg was successfully observed [81].

Accumulation of oxygen probe in specific tissues, such as kidneys and tumors, needs to be taken into consideration when computing pO_2_ values, because T_2_^*^ based linewidth estimation can be affected by self-broadening of the spin probe (trityl radical) resonance at trityl concentration greater than 5 mM [81,83]. On the other hand, T_1_-based EPR oximetry by inversion recovery electron spin echo sequence showed very low level of self-broadening effect [82,83]. A set of FID data using the saturation by fast repetition (SFR) sequence with several different TR can allow estimation of T_1_-based relaxation time and T_2_*-based relaxation time simultaneously [95].

### 2.4. Dynamic EPR Imaging (Redox Mapping)

Nitroxyl radical compounds are stable free radical species and give a characteristic EPR signal. When a nitroxyl radical is administered into live animal, it is reduced either enzymatically or chemically and loses its paramagnetic property. Nitroxyl radical loses one-electron and forms oxoammonium cation, then undergoes two-electron reduction by receiving hydrogen from in vivo hydrogen donors such as NAD(P)H and/or GSH to form hydroxylamine which is the one-electron reduced form (Figure 5). The hydroxylamine can further lose one-electron to become the nitroxyl radical (Figure 5). Such two-step reduction in nitroxyl radical mainly occurs in in vivo condition. Relatively strong reductant such as ascorbic acid can directly cause one-electron reduction and convert the nitroxyl radical to hydroxylamine (Figure 5). Due to such redox behavior, the nitroxyl radical has been employed as redox sensitive molecular probe (contrast agent) [42].

For in vivo EPR imaging of a tumor in a mouse tail data acquisition time of 32 min was required for collecting only 4 projections in the year 1987 [60]. When Alecci et al. [96] observed two sequential 2D EPR images of coronal plane of rat abdomen after administration of carboxy-PROXYL to compare the time course of image intensity, one image data set required 5 min for acquisition of 8 projections. In vivo dynamic EPR imaging was performed by several groups for following the time course of the spin probe kinetics from 1996. The advances in computer technology in 2000’s accelerated the data acquisition speed in many medical imaging modalities, including EPR imaging. In a paper reported in 2006 [97], a set of 12 projections was obtained as short as in 1.9 min/image for EPR-based redox imaging application. Fast data acquisition for a dynamic EPR imaging using carbamoyl-PROXYL as the contrast agent is available on PC based control system.

Unlike MRI, slice selection by gradient switching is not feasible in EPR imaging. To observe slice data of EPR image, which is equivalent to the MRI slice, a corresponding slice must be clipped out from a volume-rendered 3D image data. To observe time course information of a particular voxel, large number of 3D data sets must be acquired as a function of time. Hirata et al., have developed rapid scan CW EPR imaging system which is capable of repetition of 46 projections 3D image data acquisition in every 3.6 s [98].

Redox imaging using nitroxyl radicals as redox sensitive contrast agents can be performed not only by EPR imaging but also by T_1_-weighted MR imaging modality. The nitroxyl radical compounds have proton T_1_-shortening effect and can enhance T_1_-weighted MR signal, which enables high resolution redox imaging [99]. The details of MRI-based redox imaging and nitroxyl contrast agents are described elsewhere [62].

### 2.5. Co-Registration

For better understanding as well for achieving grater outcome from a functional image it is essential to relate functional information to the anatomy of the subject. OMRI scanners were used to co-register functional EPR information with the anatomy, exploiting the DNP phenomenon for NMR signal enhancement. However, such experiments needed special instrumentation. In a different approach, the EPR and NMR images were acquired in different scanners (viz; EPR imaging instrument and MRI scanner, respectively) using the same resonator without moving the subject of study, the small animal. A 300 MHz parallel-coil resonator used for the in vivo pulsed EPR imaging scanner also works on a 7 T MRI scanner because proton resonance at 7 T is also 300 MHz. Thus, the subject animal can be imaged in the 300 MHz EPR scanner as well as in the 7 T MRI scanner without changing the resonator. This way, matching anatomical images can be obtained by 7T MRI. This helps overlaying of the EPR oxygen map onto the MRI-based anatomical image [100]. Figure 6 shows oxygen maps and anatomical images of three types of pancreatic tumors acquired in a the 300 MHz pulsed EPR imager and also at 7 T MRI scanner using the same resonator without the need to remove the animal [40]. The co-registration of the oxygen maps and anatomical images enables better visualization of distribution of oxygen concentration in tumor anatomy as well as median pO_2_ and percentage of hypoxic region [101]. MRI also provides other information in the tumor such as perfusion and vasculature by using MRI contrast agents and hence this co-registration modality is also useful for investigating the relationship between vasculature/perfusion and oxygen concentration [102].

## 3. EPR Related Imaging Techniques (OMRI/PEDRI, DNP-Imaging)

### 3.1. OMRI/PEDRI

As pointed out in the introduction, the EPR-related double resonance imaging techniques take the advantage of DNP for enhancing MR signals (Figure 7). Grucker reported the first in vivo DNP imaging of nitroxyl radical in a rat abdomen [103]. The first experiment DNP-based oxygen imaging in perfused heart was reported using nitroxyl radical [104]. A combination of OMRI/PEDRI and triarylmethyl radical contrast agent provides EPR-derived pO_2_ images co-registered with MRI-based morphological information [104,105,106,107]. For in vivo OMRI/PEDRI imaging experiments, the animals are first infused with a narrow line stable free radical agent. The EPR transition is induced by irradiating the subject with RF/microwave radiation of frequency that matches with the EPR resonance frequency. The relatively large electron spin polarization can be transferred to the surrounding nuclear spins, resulting orders of magnitude enhanced NMR signal.

In principle, DNP can be realized for any nucleus of noninteger spin. The nuclei of biomedical interest include ^1^H, ^13^C, ^15^N, ^31^P, ^17^O and ^19^F. OMRI/PEDRI can provide EPR spectroscopic information based on relaxation behavior of the free radical, in addition to the highly resolved distribution of the target nucleus based on MRI acquisition [108]. The resonant magnetic field and frequency for MRI and EPRI are fairly different. In the case of a commercial OMRI system, RF radiation of 226 MHz was used for EPR excitation, corresponding to a Zeeman field of 8.1 mT, and 625 kHz was used for MR detection (Zeeman field, 15 mT) [26]. Such low field MRI is feasible largely due to the DNP enhancement. To achieve good image quality and sensitivity, a field-cycled proton-electron double-resonance imaging (FC-PEDRI) system is reported where EPR irradiation takes place at approximately 5 mT, following which the field is switched to 450 mT for NMR signal detection [109,110,111].

Application of OMRI for tumor oxymetry is vividly illustrated in Figure 8 [26]. After setting up the animal in the magnet (15–40 mT), a nontoxic and relatively stable trityl radical, OX063 (a narrow line EPR spin probe) (Figure 8A) is administered intravenously, and RF pulse is used for excitation of the EPR resonance of OX063. The electron spin polarization is transferred to surrounding proton nuclei in biological environment by dipole interaction and causes enhancement in the proton NMR signal, which is dependent on the local concentration of OX063, oxygen and on the strength of the EPR RF irradiation field. Figure 8B shows OMRI images taken at RF fields of two different magnitudes. From these two OMRI images obtained at two different EPR power levels, the spin probe OX063 distribution as well as the pO_2_ distribution (Figure 8C) can be readily computed. Figure 8C presents the increase in tumor oxygenation following carbogen breathing by the animal, illustrating the potential of this method for tumor oxymetry. In addition to pO_2_ mapping, this technique is also used for quantitative measurement of tissue permeability, redox status, and pH mapping [27,112].

### 3.2. Hyperpolarized ^13^C MRI

The DNP-based double resonance MRI techniques are referred to as hyperpolarization techniques when the nuclear polarization is dramatically increased (several orders of magnitude) as compared to the thermal equilibrium value. Hyperpolarization is achieved by performing dynamic nuclear polarization in solid state (glassy matrix) at very low temperature and at high magnetic field, to achieve maximum electron polarization and polarization transfer to the magnetic nuclei [113]. Hyperpolarized ^13^C MRI is an emerging technique offering enhanced information on metabolism, unparallel to any other current imaging techniques, with great potential for early detection of disease, staging and treatment monitoring. Further, the metabolic imaging is performed without using ionizing radiation. This method requires suitably designed ^13^C molecular probe for metabolic imaging. Stable organic radicals of very narrow single line EPR resonance are used as the source of electron spins and excited using high frequency microwave radiation. The main challenge is to bring the polarized, cold solid sample into solution without losing the polarization. The initial experiments demonstrated that high polarization, 37% for ^13^C and 7.8% for ^15^N, can be obtained in solution with corresponding signal enhancements of 44,400- and 23,500, respectively. This technique is also known by the name dissolution-DNP (*d*DNP) [113].

The schematic illustration of this technique is shown in Figure 9. A mixture of ^13^C-labeled molecular probes and OX063 in a glassy matrix in a cryomagnetic field at near 1 K is irradiated with microwave at the resonance frequency to induce excitation of unpaired electron of OX063. The electron spin polarization is transferred to nearby ^13^C nuclei in the molecular probes by thermal mixing effect. After several tens of minutes of irradiation for polarization build up, the cryogenic solid samples are rapidly dissolved in a superheated solvent to warm up around biological temperature, and then immediately released from the polarizer by positive pressure gas. The dissolved ^13^C molecular probe is then quickly subjected to an NMR tube for the acquisition of the enhanced signals. Alternatively, the samples are administered to anesthetized animals already placed in the MRI magnet to collect ^13^C MR spectra or ^13^C chemical shift image of metabolic conversion in the region of interest (ROI). Monitoring of the dynamics and chemical reactions can be performed based on the enhanced ^13^C signals with kinetic analysis [114,115].

The decay rate of hyperpolarized signal of ^13^C molecular probes is determined by the spin lattice relaxation time, T_1_ which is usually less than 1 min. To retain the hyperpolarized signals for sufficiently longer time for monitoring the crucial metabolism under physiological conditions, several modifications can be made in ^13^C-labeled molecular probe structure (e.g., removal of ^1^H to eliminate dipole–dipole interactions between ^1^H and ^13^C, enhanced target selectivity, large ^13^C chemical shift difference between the probe and metabolite products, etc.). Among the molecular probes developed, ^13^C-labeled pyruvate is so far the most promising one [116] and it is currently under clinical evaluation for prostate cancer screening and tumor response to treatment [117,118,119]. The carbonyl carbon at position 1 of the alpha-keto structure shows longer hyperpolarized lifetime (T_1_, 40–65 s) (because there are no ^1^H nuclei surrounding the carbonyl carbon). In addition, the metabolite lactate peak appears at 185 ppm, well separated from the pyruvate peak at 172 ppm in the ^13^C NMR spectrum. Moreover, pyruvate forms liquid at room temperature and glass at lower temperature. OX063 dissolves enough for hyperpolarization buildup in the liquid pyruvate. Biomedically, pyruvate is located at the intersection of oxidative phosphorylation (OXPHOS) and anaerobic fermentation. In normal tissue, pyruvate enters into mitochondrial metabolism, generating 36 ATP by OXPHOS. While in cancerous tissue or inflamed tissue, it is known that the pyruvate metabolism is shifted from OXPHOS to lactate generation in cytosol by increasing the aerobic glycolysis due to the elevated level of lactate dehydrogenase (LDH) activity known as the Warburg effect [120]. This metabolic shift is considered as one of the hallmarks of cancer and correlates with the therapeutic response. Hence, metabolic conversion of hyperpolarized ^13^C pyruvate to lactate can serve as a promising imaging biomarker. Over the past decade, we have utilized this technique to characterize the tumor metabolism, malignant transformation, and therapeutic response upon treatment with LDH inhibitors [121], hypoxia-activated prodrugs [40], and X-ray irradiation [122].

Figure 10 shows an example of HP-[1-^13^C]pyruvate metabolic imaging of mice bearing murine SCC VII or human HT29 xenograft for evaluating the effect of X-ray irradiation on tumor metabolism. The ^13^C chemical shift imaging revealed the distribution of injected ^13^C-pyruvate and the lactate formed in the tumors. The total ^13^C, and lactate to pyruvate ratio decreased after the 3 days 30 Gy irradiation, indicating decreased blood flow and lactate formation due to radiation damage whereas in the control SCCVII tumors the lactate formation increased as the tumor grew [122]. Similar trend was also observed in HT29 xenografts, in which the lactate formation decreased after irradiation. Thus, the HP-[1-^13^C]pyruvate spectroscopic imaging can be utilized for evaluating the effect of radiotherapy. Since the early report of clinical trials for evaluating prostate tumors using the molecular probe, ^13^C-pyruvate in the year 2013 [119], more than 20 of clinical hyperpolarizers (SPINlab^TM^, 5T) have been installed worldwide to study metabolism in cancer, brain, cardiovascular, kidney, liver and other diseases [123,124].

Another promising molecular probe for investigating tumor metabolism is [1,4-^13^C_2_]fumarate [125]. Fumarase catalyzes conversion of fumarate to malate in TCA cycle. Fumarase is localized inside the healthy cells, and the exogenously injected fumarate cannot effectively penetrate the healthy cell membrane. Therefore, the hyperpolarized fumarate does not get converted to malate in the image observation time frame. However, fumarase can leak from the necrotic or damaged tumor cells resulting in fumarate to malate conversion which can be detected in the observation time frame. Thus, the increase in the ^13^C signal of malate can serve as a metabolic biomarker for therapeutic response by detecting the local fumarase activity. Brindle et al. have reported the application of HP-[1,4-^13^C_2_]fumarate for detecting acute kidney injury (AKI) [126]. Recently, we have reported the applicability of fumarate for evaluating dual immune checkpoint blockade using both anti-αPD-L1 and αCTLA-4 antibodies onMC38 colon tumor model in mice (Figure 11) [127].

The development of new molecular probes is essential to further expand the utility of HP-MRS as metabolic imaging tool in cancer research. Recently, Sando et al. have reported a practical APN probe that allows in vivo detection of APN activity in tumors [128]. They designed Ala-Gly-*d*_2_-N-Me_2_ considering the physicochemical and biological conditions necessary for the HP molecular probe (Figure 12A). Using the probe, which is specifically metabolized by APN in cancer, they successfully detected the APN activity in MRS (Figure 12B–F) and CSI (Figure 12G). The probe itself has glassy characteristics similar to pyruvate and requires no additional glassing agent such as glycerol or DMSO when polarizing in the cryomagnetic field. The wide application of this probe is expected in future studies.

## 4. Possibility of Clinical Application of EPRI

EPRI has several arduousness for clinical application [14]. Two biggest obstacles for clinical application are the necessity for the approval for EPR contrast agent and large specific absorption rate (SAR). So far, no paramagnetic contrast agent for EPRI, such as nitroxides, TAM, and/or other stable free radical species, has been approved for human use. No commercialized EPR instrument for human use is available and development of EPR instrument for human and/or large animals is quite limited. Only a few groups in the world have used their own house-made instrument [7,129]. Therefore, no human application of EPRI has been reported, although several EPR spectroscopic application on human skin or other partial application has been reported from early stage of in vivo EPR studies [130,131,132]. Hyperpolarized ^13^C-MRI is simply a chemical shift imaging of an infused/injected extracorporeally hyperpolarized ^13^C-labeled compound and it’s in vivo metabolites [31]. The hyperpolarized ^13^C MRI proceeded to clinical trials after the approval for the very low dose of TAM for hyperpolarization [117,118,119]. Thus, it is theoretically feasible to get approval of clinical use for TAM when the dose for EPR is sufficiently reduced. In addition, microwave pulse power at large SAR should be reduced below certain level for safety [133]. OMRI/PEDRI, which is an MRI based imaging technique, can basically be applicable for clinical use, although it also requires large SAR of EPR excitation pulse.

In summary, application for human patients would be possible, when SAR of EPR excitation is adequately adjusted and the infusion dose of specific contrast agent, i.e., free radical compound, is approved. Technical and advantageous features of EPRI and related imaging techniques were summarized in Table 1.

## 5. Conclusions

EPRI and related imaging techniques have come a long way ever since the first EPR image of biological system was reported nearly four decades ago. EPR spectral parameters can provide information related to metabolism and physiology of tumors, such as tissue oxygenation, acidosis, redox, and glutathione status, etc. Noninvasive assessment of these parameters in animal models can be used to screen anticancer agents and optimize therapies. The sensitivity and the unique functional imaging capability for studying the tumor microenvironment to achieve these potential applications have given great impetus to research in this field. In principle, EPRI can provide maximally 5-dimensional information consisting of 3D spatial, 1D spectral, and 1D temporal variables/parameters. Development of fast switching electronic and computing devices with large on-board storage capability have tremendously contributed to multidimensional EPR imaging with high temporal resolution. Equally important contribution comes from the design and development of novel, narrow, single resonance, stable paramagnetic agents as tumor microenvironment probes. Although not yet in clinic, these developments give ample hope for future clinical applications of EPRI. Nevertheless, as a spin-off, the high sensitivity of EPR and the development of novel single resonance stable free radicals have added a new dimension to metabolic imaging using dissolution-DNP (*d*DNP). The hyperpolarized MRI, especially HP ^13^C MRI enables rapid dynamic investigation of metabolic and physiologic processes in tumors with sensitivity that was hitherto unrealizable. With novel biofunction-specific ^13^C-enriched molecular probes, carbon being the backbone of all biomolecules, HP ^13^C MRI opens the gate way for investigation of nearly all human diseases. Clinical trials evaluating the efficacy of HP ^13^C MRI are in progress. Nevertheless, there are a few major challenges. The highly sophisticated instrumentation and its cost along with the cost of ^13^C enriched molecular probes has to be addressed for the routine use of HP MRI in clinic. Complexities involved in producing the HP molecular probe, administering it to the subject and fast image acquisition require cross disciplinary expertise.

## Figures and Tables

**Figure 1 metabolites-13-00069-f001:**
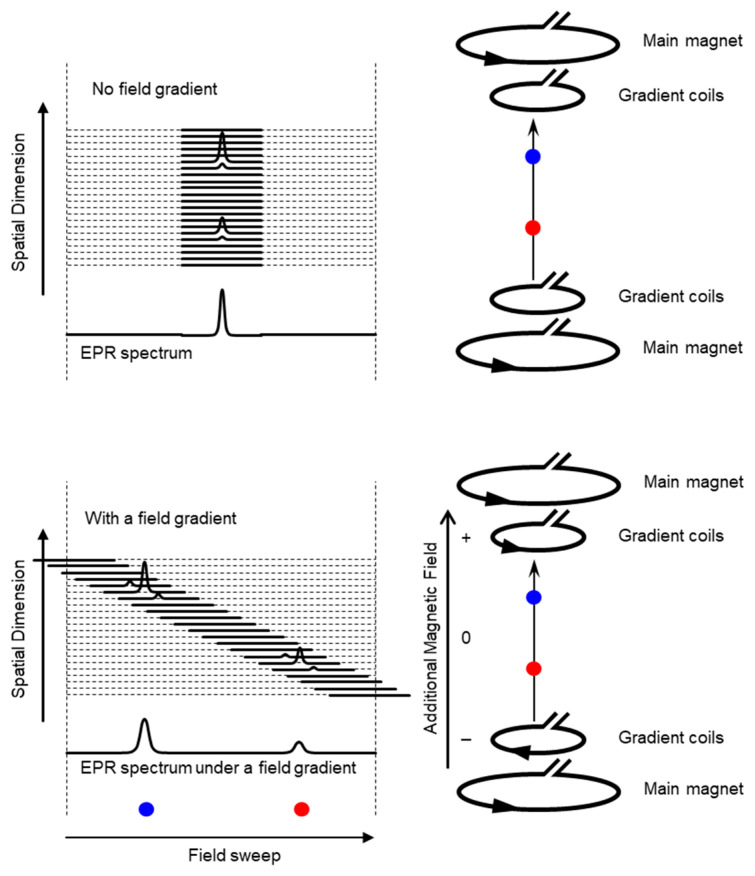
Overloading spatial information on a single resonance EPR spectrum of a free radical by magnetic field gradient. (**Upper** panel) Magnetic field is uniformly distributed in the resonator and free radical EPR is observed as a single resonance line. (**Lower** panel). When linear field gradient is applied using a pair of field gradient coils, magnetic field can be varied linearly along with the spatial position, and the single resonance EPR spectrum at different orientations of the magnetic field gradient encodes the spatial locations of the free radical (the blue and red dots) This is similar to projection reconstruction MRI first reported by Lauterbur [46]. Arrow heads on the right panels indicate direction of electric current.

**Figure 2 metabolites-13-00069-f002:**
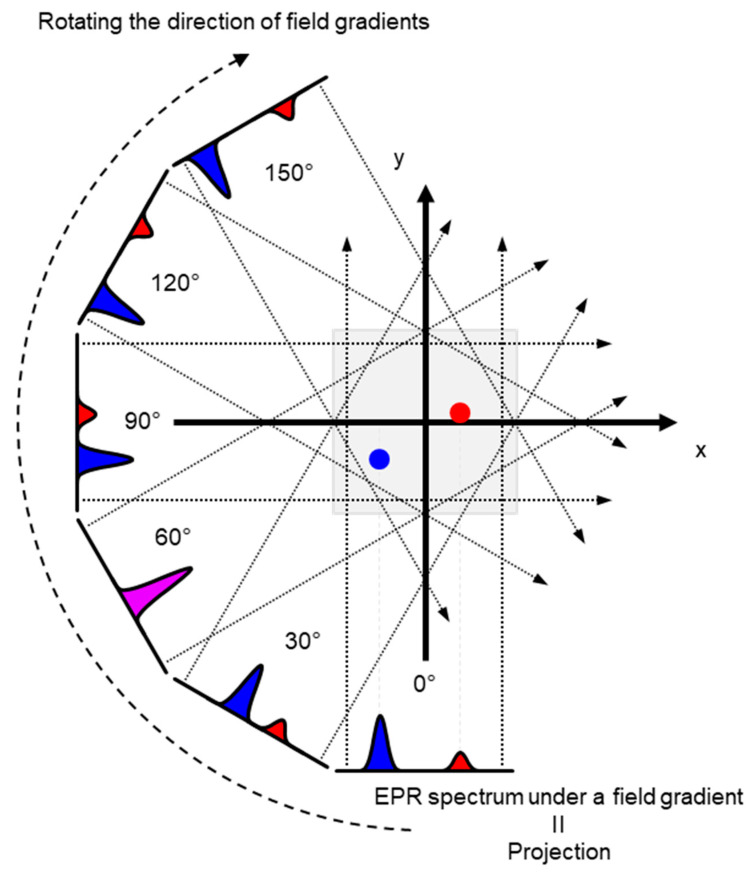
An example of 2D projection reconstruction imaging. Red and blue dots are 2D spatial positions of 2 objects. Rotating the direction of field gradient can provide different angle view of the subject, i.e., the projections. Projections are back-projected to 2D or 3D matrix, and the spatial distribution of subject can be reconstructed.

**Figure 3 metabolites-13-00069-f003:**
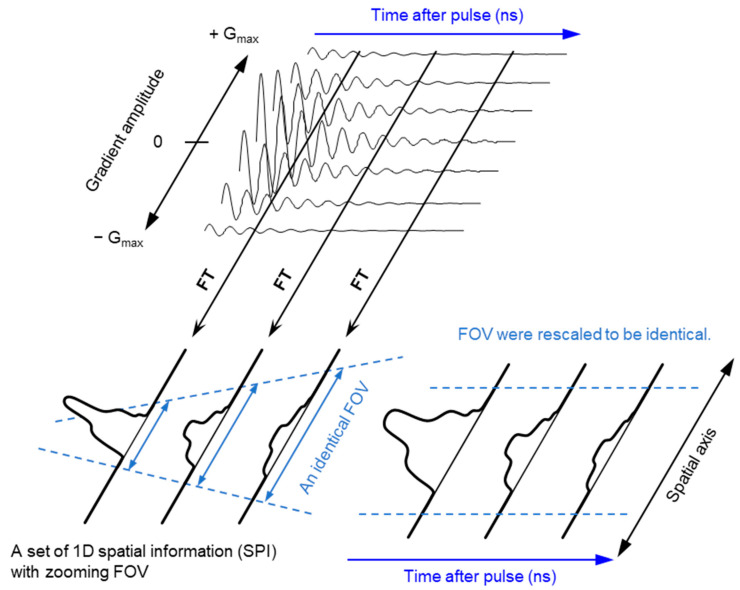
Principle of SPI and FT-CTSSI. (**Upper** panel) FID data obtained under incremented but unidirectional field gradient. This is gradient-echo-like encoding of spatial distribution along with the direction of the field gradient. (**Lower left** panel) FT of gradient amplitude axis can give 1D spatial distribution, i.e., SPI. The FOV of observed SPI is zooming in depending on the time after pulse as per the equation, FOV = 2π/(γ_e_ · τ_p_ · ΔG), where γ_e_ is the gyromagnetic ratio of the electron, τ_p_ is the time after pulse, and ΔG is the incremental step of the field gradient. (**Lower right** panel) The data in the identical FOV extracted and rescaled on a Cartesian matrix give a spectral-spatial image, i.e., 2D FT-CTSSI. FT along with the time axis can convert it to the spectra axis.

**Figure 4 metabolites-13-00069-f004:**
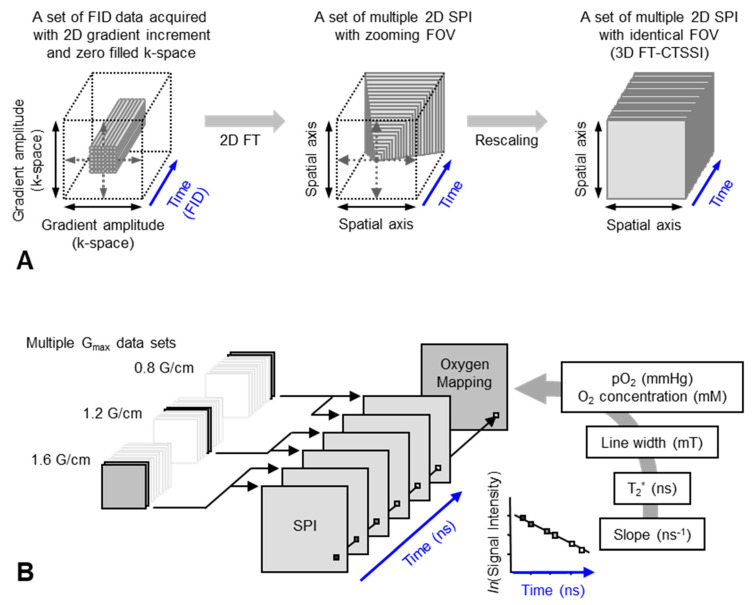
Data handling process for 2D oxygen mapping. (**A**) Data handling process for 2D SPI and 3D FT-CTSSI. Acquired FID were arranged on 3D matrix consisting of 2D k-space and 1D time axis and the k-space was zero-filled to make matrix size as 2^n^ × 2^n^. Then, the 2D k-space at every time point was 2D Fourier transformed. Two-dimensional SPI was computed for every time point, however, the FOV of SPI becomes smaller with increasing delay time. Identical FOVs were extracted and rescaled on 3D FT-CTSSI matrix, which consists of 2D spatial and 1D time axis. (Time axis was Fourier transformed to spectral axis, when required) (**B**) Correction of image resolution by multiple maximum field gradient (G_max_) data sets. Multiple CTSSI data sets were acquired with different G_max_ setting. SPI data having similar resolution were extracted from the CTSSI data sets with different G_max_ setting, and then CTSSI data set was re-assembled. T_2_* was calculated from the exponential decay of the image intensity along with time axis. T_2_* values, which are related to EPR linewidth, were converted to corresponding oxygen concentration (or pO_2_) using previously obtained standard curve.

**Figure 5 metabolites-13-00069-f005:**
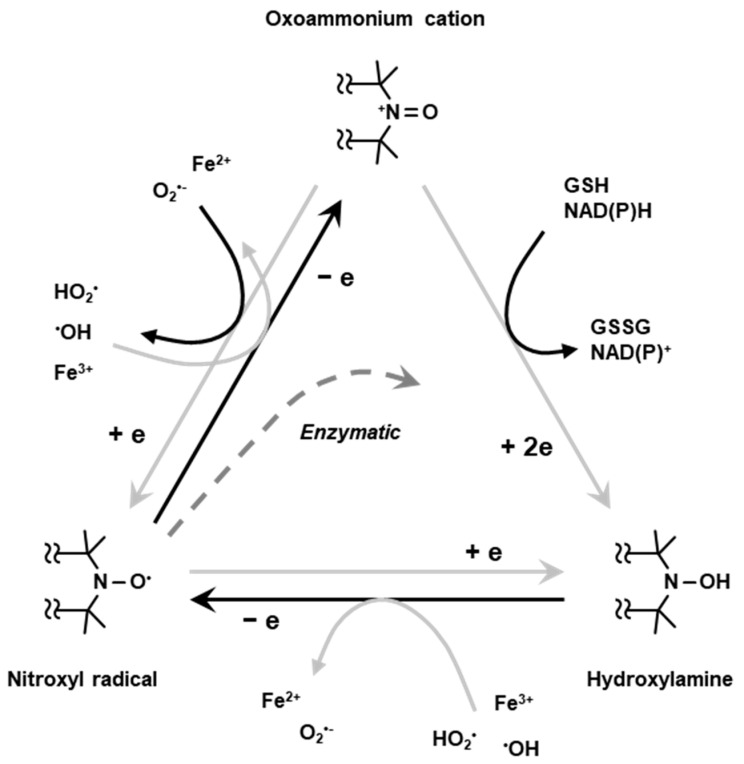
Redox cycling of nitroxyl radical. Nitroxyl radical loses one-electron chemically or enzymatically in living organisms forming corresponding oxoammonium cation. The oxoammonium cation undergoes two-electron reduction by receiving hydrogen atom from hydrogen donors, such as NAD(P)H and/or GSH, forming the corresponding hydroxylamine. The oxoammonium cation can be reduced back to be nitroxyl radical form. The hydroxylamine can lose one-electron to form nitroxyl radical again.

**Figure 6 metabolites-13-00069-f006:**
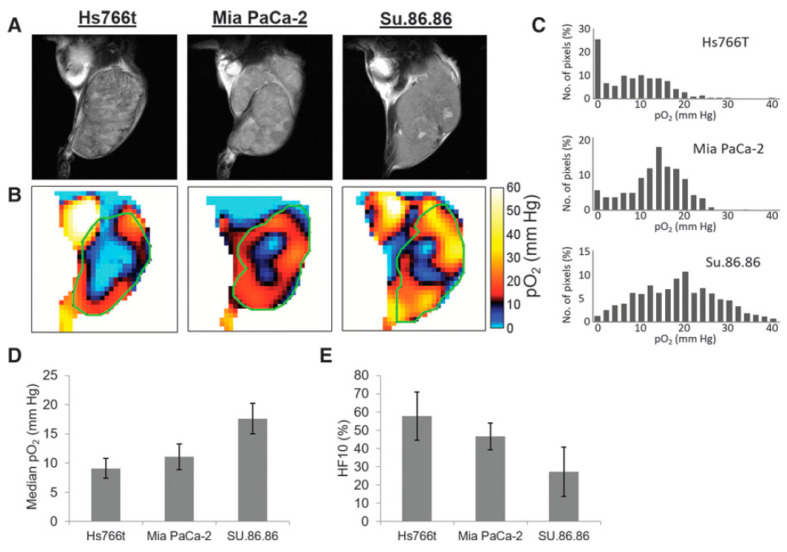
Co-registration of oxygen maps and anatomical images. (**A**), T_2_-weighted anatomical images of Hs766t, MiaPaCa2, and Su.86.86 tumors obtained by MRI. (**B**), Oxygen maps of the three tumors obtained by EPR imaging. ROI of the tumors selected from the anatomical images were overlayed for computing functional parameters. (**C**–**E**), Histograms of pO_2_ distribution in the three tumors (**C**), median pO_2_ (**D**), and hypoxic fraction with less than <10 mm Hg pO_2_ (**E**) of the pancreatic tumors.

**Figure 7 metabolites-13-00069-f007:**
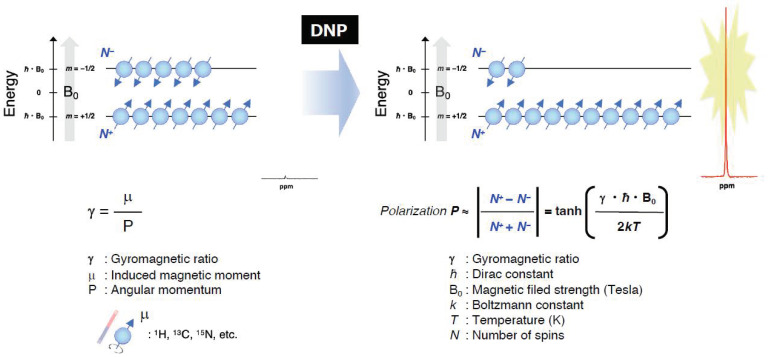
A schematic representation of the Zeeman splitting of nuclear spins (I = 1/2) and the polarization for obtaining the enhanced NMR signals by DNP under the magnetic field.

**Figure 8 metabolites-13-00069-f008:**
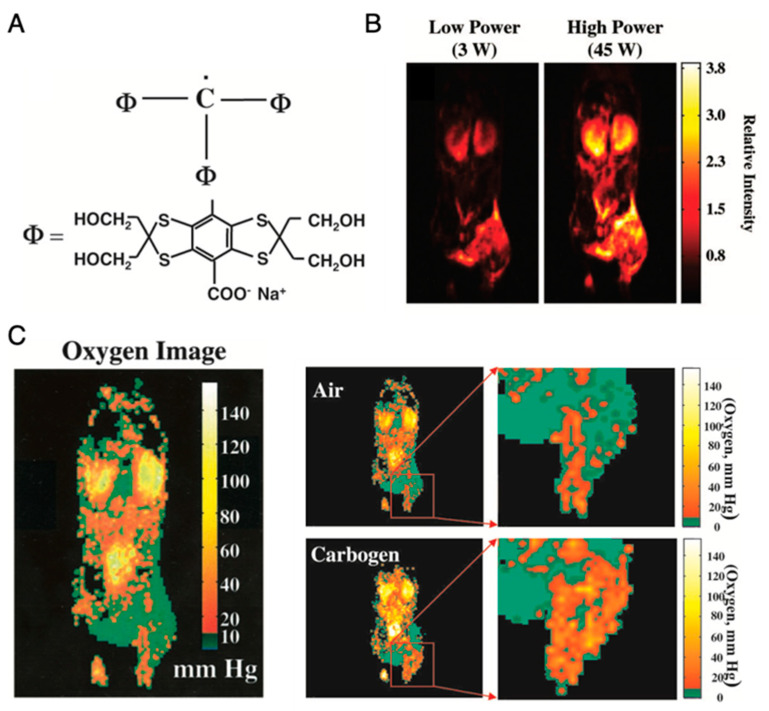
Imaging of mice pO_2_ by OMRI/PEDRI. (**A**), Structure of trimethylaryl radical OX063. (**B**), Relative intensity of the different RF power-dependent images. (**C**), Oxygen image in mice body acquired by OMRI/PEDRI. Imaging of oxygenation in the C3H mice-bearing SCCVII tumors after Carbogen challenge.

**Figure 9 metabolites-13-00069-f009:**
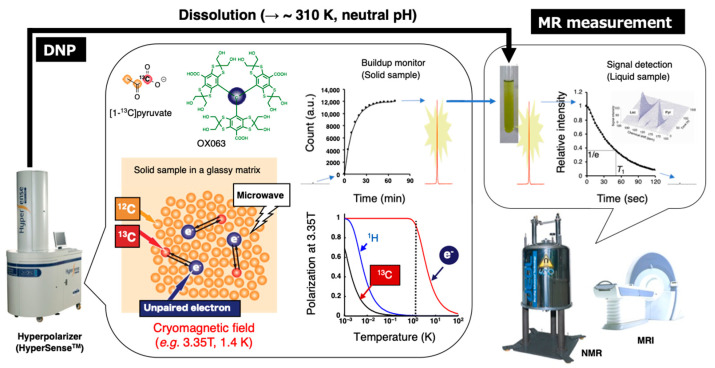
Schematic representation of hyperpolarized ^13^C NMR/MRI by dissolution DNP. The ^13^C-labeled molecular probe with the radical agent (OX063) in a glassy matrix is conditioned in the cryomagnetic field (e.g., 3.35 T, 1.4 K) on the hyperpolarizer (HyperSense^TM^). The continuous microwave irradiation preferentially excites the unpaired electron spin under the conditions and following spin transfer for the ^13^C nuclei in the molecular probe, thereby buildup the ^13^C NMR signals. The solid sample is quickly dissolved in the superheated solvent to warm up around biological temperature and pH, and then subjected to the NMR or MRI to collect the hyperpolarized spectroscopic data (HP-NMRS) or imaging (HP-MRI).

**Figure 10 metabolites-13-00069-f010:**
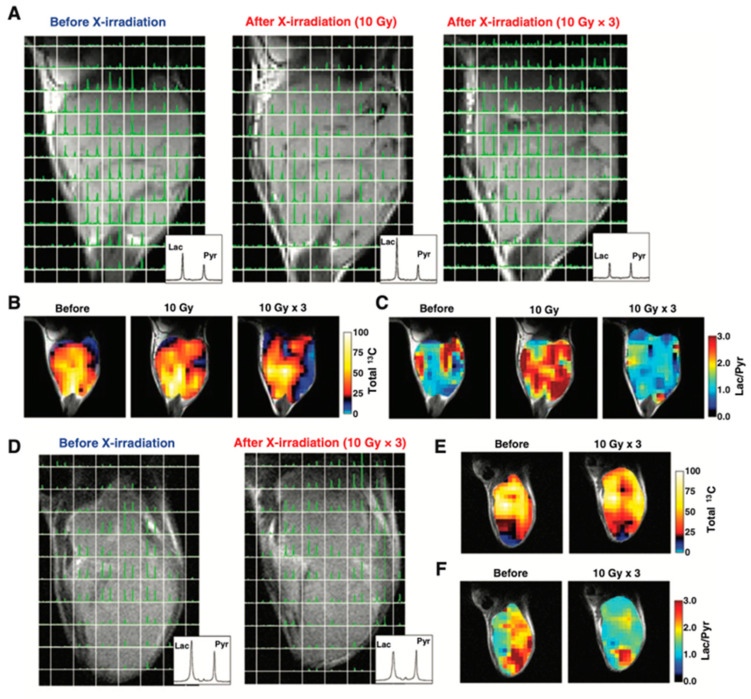
Therapeutic response of mice-bearing a SCC tumor (**A**–**C**) or a HT29 tumor (**D**–**F**) upon the X-ray irradiation. Chemical shift images of the SCC tumor (**A**) and HT29 tumor (**D**) obtained before irradiation, at 1 day after 10 Gy of irradiation, and at 1 day after fractionated 30 Gy irradiation. Total ^13^C maps (**B**,**E**) and images of the [1-^13^C]lactate to [1-^13^C]pyruvate ratio (**C**,**F**) in the SCC tumor and the HT-29 tumor calculated from the chemical shift images.

**Figure 11 metabolites-13-00069-f011:**
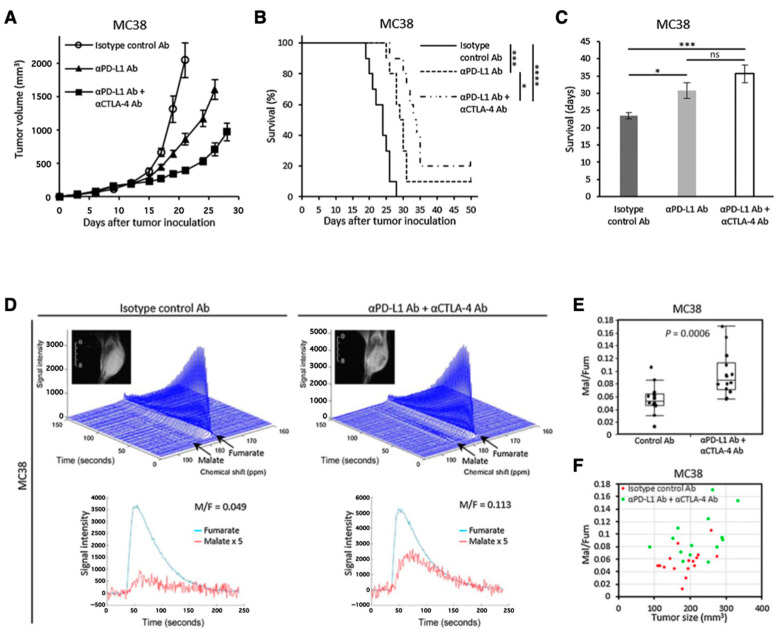
Use of [1,4-^13^C_2_]fumarate for detecting early response of tumor to immunotherapy. Growth curve (**A**) and Kaplan–Meier survival curve (**B**) of MC38 tumors inoculated in mice. The mice were treated with isotype control Ab, anti-PD-L1 Ab, or combination anti-PD-L1 Abþanti-CTLA4 Ab on days 9, 12, and 15 after the tumor inoculation. Survival refers to the time before reaching the tumor volume of 2000 mm^3^. (**C**), Bar plot of the survival from (**B**). (**D**–**F**), ^13^C-MRS of the hyperpolarized ^13^C-fumarate metabolism in the MC38 xenografts. (**D**), Representative dynamic ^13^C spectra and kinetics of fumarate and malate in the MC38 tumor treated with isotype control Ab or anti-PD-L1 Ab + anti-CTLA4 Ab. (**E**), The malate to fumarate ratio in the MC38 tumors. (**F**), Correlation between the malate to fumarate ratio and the tumor volume. * *p* < 0.05, *** *p* < 0.001 **** *p* < 0.0001. ns: non-significant differences.

**Figure 12 metabolites-13-00069-f012:**
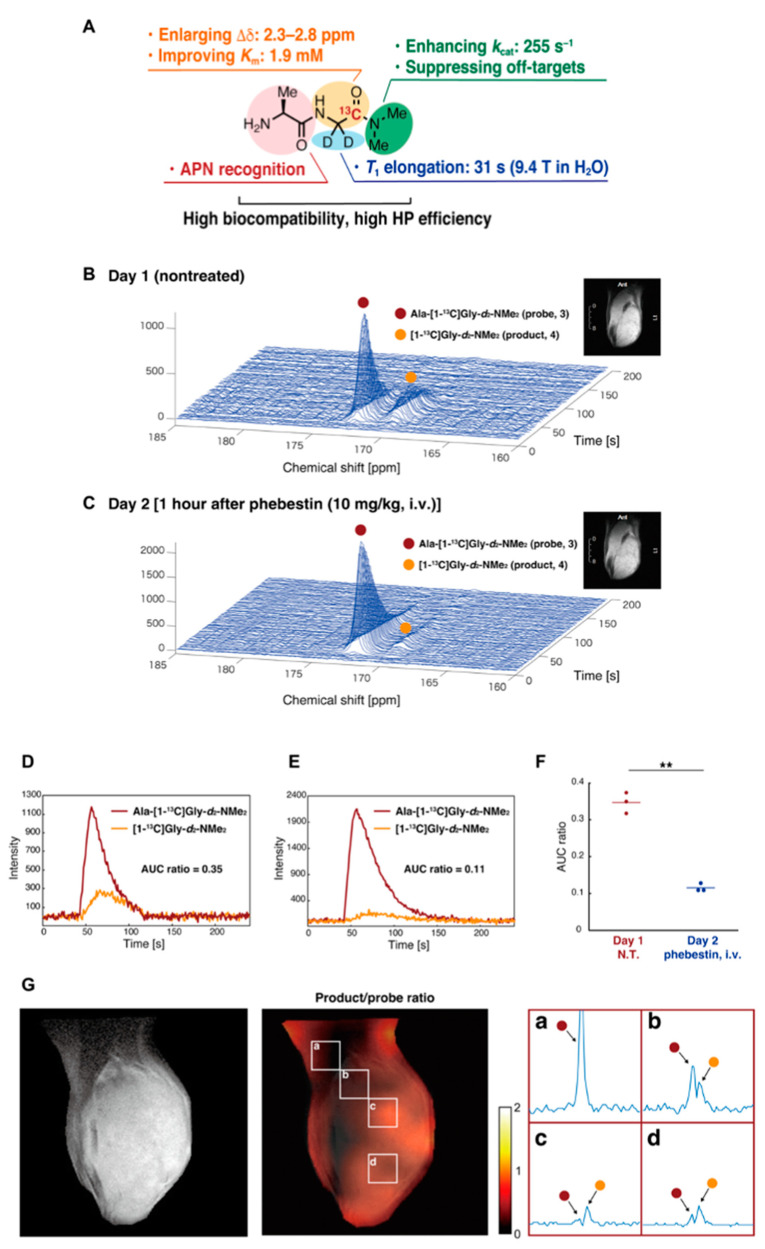
Development of a molecular probe to detect APN activity by precise molecular design. (**A**), Physicochemical and biological characteristics of the Ala-Gly-d_2_-N-Me_2_. (**B**), Detection of the APN activity in mice-bearing MiaPaCa-2 tumor xenografts. Nontreated (**B**) treated (**C**) with the APN inhibitor (1 h after the phebestion administration) in the next day. (**D**–**F**), The profile of APN probe and the metabolites nontreated (**D**) or treated (**E**) with phebestin. The data are quantitatively shown as AUC ratios. (**G**), Co-registration of T2-weighted anatomical image and ^13^C CSI image of the APN probe. The part of the matrix (a–d) is expanded. **: n = 3 and *P* = 0.009.

**Table 1 metabolites-13-00069-t001:** Technical and practical features of EPRI and related imaging techniques.

Modality	Application	Observation Object	Advantageous Feature	Reference	Possibility of Clinical Application
CW EPRI	Distribution mapping	Signal intensity	Direct and quantitative detection of free radical species	[41,61,62,63,64,96,97,98]	Applying EPR to whole or stem of human would be difficult because of lower intensity due to lower RF frequency worked on larger resonator [14]. However, a limited partial application would be available for human [130,131,132].
	Redox mapping	Signal decay rate	Reduction rate observed based on the direct and quantitative detection of free radical species	[41,97,98]	
	Oxygen mapping	EPR linewidth (Relaxation time)	Wide range of free radical species as O_2_-probe.	[88,89,94]	
		Signal intensity loss by RF power saturation (Relaxation behavior)	Simple acquisition process with only two images observed under different RF power	[67]	
	Separately mapping multiple free radical species	Difference of EPR resonant field	Wide spectral window has wide applicability.	[90,91]	
Pulsed EPRI	Distribution mapping	Signal intensity	Rapid acquisition is available.	[74,77,78,79]	
	Oxygen mapping	T_2_^*^ and/or T_1_ relaxation time	Quantitative and high resolution O_2_-mapping	[19,25,40,75,76,80,81,95,100,101,102,134,135,136]	
OMRI/PEDRI	Oxygen mapping	Electron relaxation	Quantitative and high resolution O_2_-mapping	[26,103,104,105,106,107,108,112]	Application for human would be possible, when SAR of EPR excitation was reduced accordingly, and when the free radical compound used for hyperpolarization was approved [14].
	Redox mapping	Signal decay rate	High spatial and temporal resolution, and slice selection	[28,29,30,43,108]	
	Separately mapping multiple free radical species	Difference of EPR resonant field	High spatial and temporal resolution, and slice selection	[27,108]	
Hyperpolarized ^13^C MRI	Mapping metabolic shift	Chemical shift of ^13^C-labeled compounds	Extracorporeally hyperpolarized ^13^C-labeled compound	[25,40,31,113,114,115,116,117,118,119,121,122,123,124,125,126,127,128]	Application for human patients has been reported [117,118,119].

* Reference numbers in parenthesis have applied the corresponding imaging modality in multimodal/preclinical/clinical investigations.

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
