# Peer review of "EPR and Related Magnetic Resonance Imaging Techniques in Cancer Research"

_metabolites, 2023, doi:10.3390/metabo13010069_

Round 1

Reviewer 1 Report

This paper is a review of EPR and related imaging techniques. The manuscript is well documented, and the authors did much work to review from the development to the utility of EPR. I have just some minor comments:

- The manuscript needs a little bit of organization. I suggest adding a Summary table, including the diverse imaging techniques you described in the manuscript, to help the readers get an overview and compare each technique.

- EPR should be addressed in full term once again, although it has been in the abstract. Also, DNP.

- Side effects and clinical application of EPRI are described as somewhat insufficient. It would be better to add some more reported studies on the clinical application of these techniques or suggest potential clinical applications.

Reviewer 2 Report

The current manuscript is generally well organized and comprehensive. I only have one suggestion. I'd recommend the authors to include a table to compare the pros and cons of different EPR relevant techniques to make it more readily understandable. 

Reviewer 3 Report

Authors present a review article on imaging strategies mapping physiologic and metabolic aspects in vivo  within the framework of their application in cancer research; Electron Paramagnetic Resonance Imaging (EPRI) for pO2 assessment in tumor microenvironment as well as Dynamic nuclear polarization (DNP) of 13C nuclei have been analyzed. As we understand from informative introduction, at least one of these techniques (EPRI) is used only in animal studies and in experimental setting.

The authors should state in their Methodology is this a narrative, a scoping or contemporary review and what were the criteria for selection of manuscripts. The manuscript is full with technical aspects and explanations regarding the process of imaging and processing, however clinical use is unclear. What manuscripts lacks is a subsection dedicated to clinical applications of these imaging modalities with illustrative cases and examples. In Lines 106-132 authors elaborate the use for radiation therapy planning, however concrete examples are welcome in order to provide more scientific input from this review.  This should specifically correlate to oncology, since it is written in very generalized way without proper accentuation on application for cancer. 

Reviewer 4 Report

The authors Takakusagi et al wrote a systematic review on “EPR and related magnetic resonance imaging techniques in 2 cancer research. The authors updated the current knowledge on imaging tumor microenvironments by strategies mapping physiologic and metabolic aspects in vivo and their application in cancer research, highlighting each of their potential and challenges. The subject is novel and well-written. I suggest authors address the following comments.

Major Comments

1.     I suggest authors discuss the advantages and limitations of the new imaging modalities in clinical trials.

2.     I would suggest authors correct a few typos and grammatical errors wherever applicable.

Round 2

Reviewer 3 Report

Authors have sufficiently responded to reviewers remarks, as for the clinical aspect of the manuscript I suggest to publish. Authors have sufficiently responded to reviewers remarks, as for the clinical aspect of the manuscript I suggest to publish. Technical part should be reviewed by MRI physicist.